# Using Trusted Data to Train Deep Networks on Labels Corrupted by Severe Noise

**Dan Hendrycks**[*]
University of California, Berkeley
hendrycks@berkeley.edu

**Mantas Mazeika**[*]
University of Chicago
mantas@ttic.edu

**Duncan Wilson**
Foundational Research Institute
duncanw@nevada.unr.edu

**Kevin Gimpel**
Toyota Technological Institute at Chicago
kgimpel@ttic.edu

## Abstract

The growing importance of massive datasets used for deep learning makes robustness to label noise a critical property for classifiers to have. Sources of label noise include automatic labeling, non-expert labeling, and label corruption by data poisoning adversaries. Numerous previous works assume that no source of labels can be trusted. We relax this assumption and assume that a small subset of the training data is trusted. This enables substantial label corruption robustness performance gains. In addition, particularly severe label noise can be combated by using a set of trusted data with clean labels. We utilize trusted data by proposing a loss correction technique that utilizes trusted examples in a data-efficient manner to mitigate the effects of label noise on deep neural network classifiers. Across vision and natural language processing tasks, we experiment with various label noises at several strengths, and show that our method significantly outperforms existing methods.

## 1 Introduction

Robustness to label noise is set to become an increasingly important property of supervised learning models. With the advent of deep learning, the need for more labeled data makes it inevitable that not all examples will have high-quality labels. This is especially true of data sources that admit automatic label extraction, such as web crawling for images, and tasks for which high-quality labels are expensive to produce, such as semantic segmentation or parsing. Additionally, label corruption may arise in data poisoning [10, 24]. Both natural and malicious label corruptions tend to sharply degrade the performance of classification systems [30].

Most prior work on label corruption robustness assumes that all training data are potentially corrupted. However, it is usually the case that a number of trusted examples are available. Trusted data are gathered to create validation and test sets. When it is possible to curate trusted data, a small set of trusted data could be created for training. We depart from the assumption that all training data are potentially corrupted by assuming that a subset of the training is trusted. In turn we demonstrate that having some amount of trusted training data enables significant robustness gains.

To leverage the additional information from trusted labels, we propose a new loss correction and empirically verify it on a number of vision and natural language datasets with label corruption. Specifically, we demonstrate recovery from extremely high levels of label noise, including the dire case when the untrusted data has a majority of its labels corrupted. Such severe corruption can occur in adversarial situations like data poisoning, or when the number of classes is large. In comparison to

---

[*]Equal contribution.

loss corrections that do not employ trusted data [18], our method is significantly more accurate in problem settings with moderate to severe label noise. Relative to a recent method which also uses trusted data [11], our method is far more data-efficient and generally more accurate. These results demonstrate that systems can weather label corruption with access only to a small number of gold standard labels. Experiment code is available at https://github.com/mmazeika/glc.

## 2   Related Work

The performance of machine learning systems reliant on labeled data has been shown to degrade noticeably in the presence of label noise [17, 19]. In the case of adversarial label noise, this degradation can be even worse [20]. Accordingly, modeling, correcting, and learning with noisy labels has been well studied [16, 1, 3].

The methods of [15, 9, 18, 25] allow for label noise robustness by modifying the model's architecture or by implementing a loss correction. Unlike Mnih and Hinton [15] who focus on binary classification of aerial images and Larsen et al. [9] who assume symmetric label noise, [18, 25] consider label noise in the multi-class problem setting with asymmetric noise.

Sukhbaatar et al. [25] introduce a stochastic matrix measuring label corruption, note its inability to be calculated without access to the true labels, and propose a method of forward loss correction. Forward loss correction adds a linear layer to the end of the model and the loss is adjusted accordingly to incorporate learning about the label noise. Patrini et al. [18] also make use of the forward loss correction mechanism, and propose an estimate of the label corruption estimation matrix which relies on strong assumptions, and does not make use of clean labels.

Contra [25, 18], we make the assumption that during training the model has access to a small set of clean labels. See Charikar, Steinhardt, and Valiant [2] for a general analysis of this assumption. This assumption has been leveraged by others for the purpose of label noise robustness, most notably [26, 11, 27, 21]. Veit et al. [26] use human-verified labels to train a label cleaning network by estimating the residuals between the noisy and clean labels in a multi-label classification setting. In the multi-class setting that we focus on in this work, Li et al. [11] propose distilling the predictions of a model trained on clean labels into a second network trained on these predictions and the noisy labels. Our work differs from these two in that we do not train neural networks on the clean labels alone.

## 3   Gold Loss Correction

We are given an untrusted dataset $\widetilde{\mathcal{D}}$ of $u$ examples $(x, \widetilde{y})$, and we assume that these examples are *potentially* corrupted examples from the true data distribution $p(x, y)$ with $K$ classes. Corruption is specified by a label noise distribution $p(\widetilde{y} \mid y, x)$. We are also given a trusted dataset $\mathcal{D}$ of $t$ examples drawn from $p(x, y)$, where $t/u \ll 1$. We refer to $t/u$ as the trusted fraction. Concretely, a web scraper labeling images from metadata may produce an untrusted set, while expert-annotated examples would form a trusted dataset and be a *gold standard*.

We explore two avenues of utilizing $\mathcal{D}$ to improve this approach. The first directly uses the trusted data while training the final classifier. As this could be applied to existing methods, we run ablations to demonstrate its effect. The second avenue uses the additional information conferred by the clean labels to better model the label noise for use in a corrected classifier.

Our method makes use of $\mathcal{D}$ to estimate the $K \times K$ matrix of corruption probabilities $C_{ij} = p(\widetilde{y} = j \mid y = i)$. Once this estimate is obtained, we use it to train a modified classifier from which we recover an estimate of the desired conditional distribution $p(y \mid x)$. We call this method the Gold Loss Correction (GLC), so named because we make use of trusted or gold standard labels.

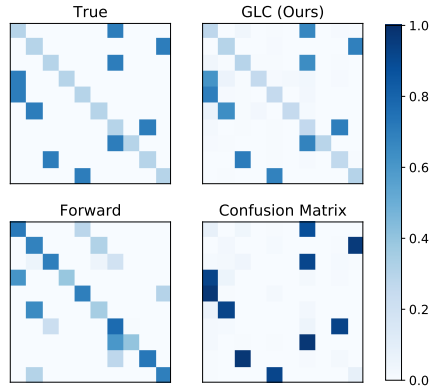

Figure 1: A label corruption matrix (top left) and three matrix estimates for a corrupted CIFAR-10 dataset. Entry $C_{ij}$ is the probability that a label of class $i$ is corrupted to class $j$, or symbolically $C_{ij} = p(\widetilde{y} = j|y = i)$.

**Estimating The Corruption Matrix.**    First, we train a classifier $\widehat{p}(\widetilde{y} \mid x)$ on $\widetilde{\mathcal{D}}$. Let $\widetilde{y}$ and $y$ be in the set of possible labels. To estimate the probability $p(\widetilde{y} \mid y)$, we use the identity $p(\widetilde{y} \mid y, x)p(x \mid y) = p(\widetilde{y} \mid y)p(x \mid \widetilde{y}, y)$. Integrating over all $x$ gives us

$$\int p(\widetilde{y} \mid y, x)p(x \mid y)\,\mathrm{d}x = p(\widetilde{y} \mid y)\int p(x \mid \widetilde{y}, y)\,\mathrm{d}x = p(\widetilde{y} \mid y).$$

We can approximate the integral on the left with the expectation of $p(\widetilde{y} \mid y, x)$ over the empirical distribution of $x$ given $y$. Assuming conditional independence of $\widetilde{y}$ and $y$ given $x$, we have $p(\widetilde{y} \mid y, x) = p(\widetilde{y} \mid x)$, which is directly approximated by $\widehat{p}(\widetilde{y} \mid x)$, the classifier trained on $\widetilde{\mathcal{D}}$. More explicitly, let $A_i$ be the subset of $x$ in $\mathcal{D}$ with label $i$. Denote our estimate of $C$ by $\widehat{C}$. We have

$$\widehat{C}_{ij} = \frac{1}{|A_i|}\sum_{x \in A_i} \widehat{p}(\widetilde{y} = j \mid x) = \frac{1}{|A_i|}\sum_{x \in A_i} \widehat{p}(\widetilde{y} = j \mid y = i, x) \approx p(\widetilde{y} = j \mid y = i).$$

This is how we estimate our corruption matrix for the GLC. The approximation relies on $\widehat{p}(\widetilde{y} \mid x)$ being a good estimate of $p(\widetilde{y} \mid x)$, on the number of trusted examples of each class, and on the extent to which the conditional independence assumption is satisfied. The conditional independence assumption is reasonable, as it is usually the case that noisy labeling processes do not have access to the true label. Moreover, when the data are separable (i.e. $y$ is deterministic given $x$), the assumption follows. A proof of this is provided in the Supplementary Material. We investigate these factors in experiments.

**Training a Corrected Classifier.**

Now with $\widehat{C}$, we follow the method of [25, 18] to train a corrected classifier, which we now briefly describe. Given the $K \times 1$ softmax output $s$ of an untrained classifier, we define the new output as $\widetilde{s} := \widehat{C}^{\mathsf{T}}s$. We then train $\widehat{p}(\widetilde{s} \mid x)$ on the noisy labels with cross-entropy loss. We can further improve on this method by using trusted data to train the corrected classifier. Thus, we apply no correction on examples from the trusted set encountered during training. This has the effect of allowing the GLC to handle a degree of instance-dependency in the label noise [14], though our experiments suggest that most of the GLC's performance gains derive from our $\widehat{C}$ estimate. A concrete algorithm of our method is provided here.

---

**Algorithm** GOLD LOSS CORRECTION (GLC)

1: **Input:** Trusted data $\mathcal{D}$, untrusted data $\widetilde{\mathcal{D}}$, loss $\ell$
2: Train network $f(x) = \widehat{p}(\widetilde{y}|x; \theta) \in \mathbb{R}^K$ on $\widetilde{\mathcal{D}}$
3: Fill $\widehat{C} \in \mathbb{R}^{K \times K}$ with zeros
4: **for** $k = 1, \ldots, K$ **do**
5:     $\texttt{num\_examples} = 0$
6:     **for** $(x_i, y_i) \in \mathcal{D}$ such that $y_i = k$ **do**
7:         $\texttt{num\_examples} \mathrel{+}= 1$
8:         $\widehat{C}_{k\bullet} \mathrel{+}= f(x_i)$ {add $f(x_i)$ to $k$th row}
9:     **end for**
10:     $\widehat{C}_{k\bullet} \mathrel{/}= \texttt{num\_examples}$
11: **end for**
12: Initialize new model $g(x) = \widehat{p}(y|x; \theta)$
13: Train with $\ell(g(x), y)$ on $\mathcal{D}$, $\ell(\widehat{C}^{\mathsf{T}}g(x), \widetilde{y})$ on $\widetilde{\mathcal{D}}$
14: **Output:** Model $\widehat{p}(y|x; \theta)$

---

## 4   Experiments

**Generating Corrupted Labels.**    Suppose our dataset has $t + u$ examples. We sample a set of $t$ trusted datapoints $\mathcal{D}$, and the remaining $u$ untrusted examples form $\widetilde{\mathcal{D}}$, which we probabilistically corrupt according to a true corruption matrix $C$. Note that we do not have knowledge of which of our $u$ untrusted examples are corrupted. We only know that they are *potentially* corrupted.

To generate the untrusted labels from the true labels in $\widetilde{\mathcal{D}}$, we first obtain a corruption matrix $C$. Then, for an example with true label $i$, we sample the corrupted label from the categorical distribution parameterized by the $i$th row of $C$. Note that this does not satisfy the conditional independence assumption required for our estimate of $C$. However, we find that the GLC still works well in practice, perhaps because this assumption is also satisfied when the data are separable, in the sense that each $x$ has a single true $y$, which is approximately true in many of our experiments.

**Comparing Loss Correction Methods.**    The GLC differs from previous loss corrections for label noise in that it reasonably assumes access to a high-quality annotation source. Therefore, to compare to other loss correction methods, we ask how each method performs when starting from the same dataset with the same label noise. In other words, the only additional information our method uses is knowledge of which examples are trusted, and which are potentially corrupted.

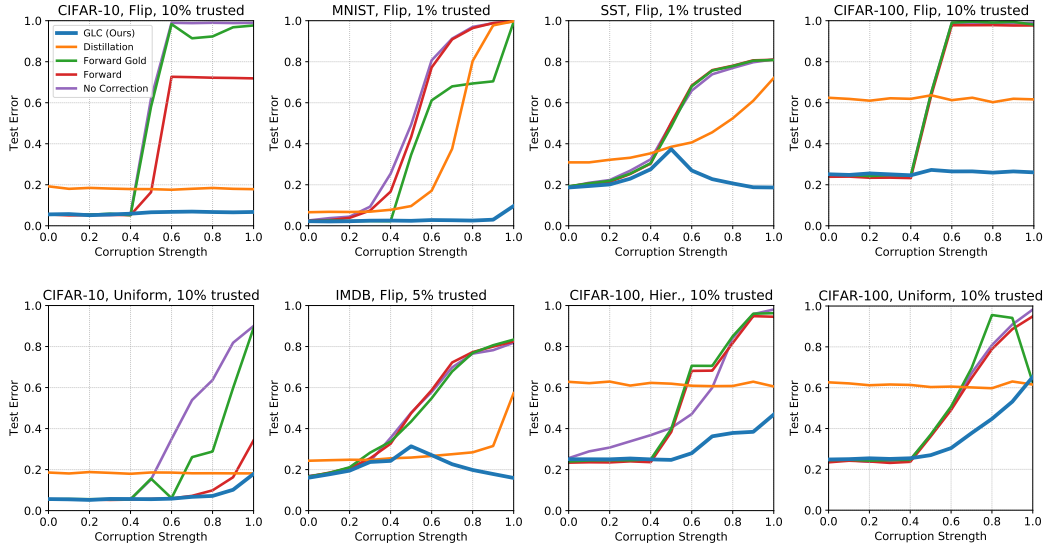

Figure 2: Error curves for numerous label correction methods using a full range of label corruption strengths on several different vision and natural language processing datasets.

## 4.1 Datasets and Architectures

**MNIST.** The MNIST dataset contains $28 \times 28$ grayscale images of the digits 0-9. The training set has 60,000 images and the test set has 10,000 images. For preprocessing, we rescale the pixels to the interval $[0, 1]$. We train a 2-layer fully connected network with 256 hidden dimensions. We train with Adam for 10 epochs using batches of size 32 and a learning rate of 0.001. For regularization, we use $\ell_2$ weight decay on all layers with $\lambda = 1 \times 10^{-6}$.

**CIFAR.** The two CIFAR datasets contain $32 \times 32 \times 3$ color images. CIFAR-10 has ten classes, and CIFAR-100 has 100 classes. CIFAR-100 has 20 "superclasses" which partition its 100 classes into 20 semantically similar sets. We use these superclasses for hierarchical noise. Both datasets have 50,000 training images and 10,000 testing images. For both datasets, we train a Wide Residual Network [29] of depth 40 and a widening factor of 2. We train for 75 epochs using SGD with Nesterov momentum and a cosine learning rate schedule [12].

**IMDB.** The IMDB Large Movie Reviews dataset [13] contains 50,000 highly polarized movie reviews from the Internet Movie Database, split evenly into train and test sets. We pad and clip reviews to a length of 200 tokens, and learn 50-dimensional word vectors from scratch for a vocab size of 5,000. We train an LSTM with 64 hidden dimensions on this data. We train using the Adam optimizer [8] for 3 epochs with batch size 64 and the suggested learning rate of 0.001. For regularization, we use dropout [23] on the linear output layer with a dropping probability of 0.2.

**Twitter.** The Twitter Part of Speech dataset [4] contains 1,827 tweets annotated with 25 POS tags. This is split into a training set of 1,000 tweets, a development set of 327 tweets, and a test set of 500 tweets. We use the development set to augment the training set. We use pretrained 50-dimensional word vectors, and for each token, we concatenate word vectors in a fixed window centered on the token. These form our training and test set. We use a window size of 3, and train a 2-layer fully connected network with hidden size 256, and use the GELU nonlinearity [7]. We train with Adam for 15 epochs with batch size 64 and learning rate of 0.001. For regularization, we use $\ell_2$ weight decay with $\lambda = 5 \times 10^{-5}$ on all but the linear output layer.

**SST.** The Stanford Sentiment Treebank dataset consists of single sentence movie reviews [22]. We use the 2-class version (i.e. SST2), which has 6,911 reviews in the training set, 872 in the development set, and 1,821 in the test set. We use the development set to augment the training set. We pad and clip reviews to a length of 30 tokens and learn 100-dimensional word vectors from scratch for a vocab size of 10,000. Our classifier is a word-averaging model with an affine output layer. We use the Adam optimizer for 5 epochs with batch size 50 and learning rate 0.001. For regularization, we use $\ell_2$ weight decay with $\lambda = 1 \times 10^{-4}$ on the output layer.

| | Corruption Type | Percent Trusted | Trusted Only | No Corr. | Forward | Forward Gold | Distill. | Confusion Matrix | GLC (Ours) |
|---|---|---|---|---|---|---|---|---|---|
| MNIST | Uniform | 5 | 37.6 | 12.9 | 14.5 | 13.5 | 42.1 | 21.8 | 10.3 |
| | Uniform | 10 | 12.9 | 12.3 | 13.9 | 12.3 | 9.2 | 15.1 | 6.3 |
| | Uniform | 25 | 6.6 | 9.3 | 11.8 | 9.2 | 5.8 | 11.0 | 4.7 |
| | Flip | 5 | 37.6 | 50.1 | 51.7 | 41.4 | 46.5 | 11.7 | 3.4 |
| | Flip | 10 | 12.9 | 51.1 | 48.8 | 36.4 | 32.4 | 5.6 | 2.9 |
| | Flip | 25 | 6.6 | 47.7 | 50.2 | 37.1 | 28.2 | 3.8 | 2.6 |
| | Mean | | 19.0 | 30.6 | 31.8 | 25.0 | 27.4 | 11.5 | **5.0** |
| CIFAR-10 | Uniform | 5 | 39.6 | 31.9 | 9.1 | 27.8 | 29.7 | 22.4 | 9.0 |
| | Uniform | 10 | 31.3 | 31.9 | 8.6 | 20.6 | 18.3 | 22.7 | 6.9 |
| | Uniform | 25 | 17.4 | 32.7 | 7.7 | 27.1 | 11.6 | 16.7 | 6.4 |
| | Flip | 5 | 39.6 | 53.3 | 38.6 | 47.8 | 29.7 | 8.1 | 6.6 |
| | Flip | 10 | 31.3 | 53.2 | 36.5 | 51.0 | 18.1 | 8.2 | 6.2 |
| | Flip | 25 | 17.4 | 52.7 | 37.6 | 49.5 | 11.8 | 7.1 | 6.1 |
| | Mean | | 29.4 | 42.6 | 23.0 | 37.3 | 19.9 | 14.2 | **6.9** |
| CIFAR-100 | Uniform | 5 | 82.4 | 48.8 | 47.7 | 49.6 | 87.5 | 53.6 | 42.4 |
| | Uniform | 10 | 67.3 | 48.4 | 47.2 | 48.9 | 61.2 | 49.7 | 33.9 |
| | Uniform | 25 | 52.2 | 45.4 | 43.6 | 46.0 | 39.8 | 39.6 | 27.3 |
| | Flip | 5 | 82.4 | 62.1 | 61.6 | 62.6 | 87.1 | 28.6 | 27.1 |
| | Flip | 10 | 67.3 | 61.9 | 61.0 | 62.2 | 61.8 | 26.9 | 25.8 |
| | Flip | 25 | 52.2 | 59.6 | 57.5 | 61.4 | 40.0 | 25.1 | 24.7 |
| | Hierarchical | 5 | 82.4 | 50.9 | 51.0 | 52.4 | 87.1 | 45.8 | 34.8 |
| | Hierarchical | 10 | 67.3 | 51.9 | 50.5 | 52.1 | 61.7 | 38.8 | 30.2 |
| | Hierarchical | 25 | 52.2 | 54.3 | 47.0 | 51.1 | 39.7 | 29.7 | 25.4 |
| | Mean | | 67.3 | 53.7 | 51.9 | 54.0 | 62.9 | 37.5 | **30.2** |

Table 1: Vision dataset results. Percent trusted is the trusted fraction multiplied by 100. Unless otherwise indicated, all values are percentages representing the area under the error curve computed at 11 test points. The best mean result is bolded.

## 4.2 Label Noise Corrections

**Forward Loss Correction.** The forward correction method from Patrini et al. [18] also obtains $\widehat{C}$ by training a classifier on the noisy labels, and using the resulting softmax probabilities. However, this method does not make use of a trusted fraction of the training data. Instead, it uses the argmax at the 97[th] percentile of softmax probabilities for a given class as a heuristic for detecting an example that is truly a member of said class. As in the original paper, we replace this with the argmax over all softmax probabilities for a given class on CIFAR-100 experiments. The estimate of $C$ is then used to train a corrected classifier in the same way as the GLC.

**Forward Gold.** To examine the effect of training on trusted labels as done by the GLC, we augment the Forward method by replacing its estimate of $C$ with the identity on trusted examples. We call this method Forward Gold. It can also be seen as the GLC with the Forward method's estimate of $C$.

**Distillation.** The distillation method of Li et al. [11] involves training a neural network on a large trusted dataset and using this network to provide soft targets for the untrusted data. In this way, labels are "distilled" from a neural network. If the classifier's decisions for untrusted inputs are less reliable than the original noisy labels, then the network's utility is limited. Thus, to obtain a reliable neural network, a large trusted dataset is necessary. A new classifier is trained using labels that are a convex combination of the soft targets and the original untrusted labels.

**Confusion Matrices.** An intuitive alternative to the GLC is to estimate $C$ by a confusion matrix. To do this, we train a classifier on the untrusted examples, obtain its confusion matrix on the trusted examples, row-normalize the matrix, and then train a corrected classifier as in the GLC.

## 4.3 Uniform, Flip, and Hierarchical Corruption

**Corruption-Generating Matrices.** We consider three types of corruption matrices: corrupting uniformly to all classes, i.e. $C_{ij} = 1/K$, flipping a label to a different class, and corrupting uniformly to classes which are semantically similar. To create a uniform corruption at different strengths, we

| | Corruption Type | Percent Trusted | Trusted Only | No Corr. | Forward | Forward Gold | Distill. | Confusion Matrix | GLC (Ours) |
|---|---|---|---|---|---|---|---|---|---|
| SST | Uniform | 5 | 45.4 | 27.5 | 26.5 | 26.6 | 43.4 | 26.1 | 24.2 |
| | Uniform | 10 | 35.2 | 27.2 | 26.2 | 25.9 | 33.3 | 25.0 | 23.5 |
| | Uniform | 25 | 26.1 | 26.5 | 25.3 | 24.6 | 25.0 | 22.4 | 21.7 |
| | Flip | 5 | 45.4 | 50.2 | 50.3 | 50.3 | 48.8 | 26.0 | 24.9 |
| | Flip | 10 | 35.2 | 49.9 | 50.1 | 49.9 | 42.1 | 24.6 | 23.5 |
| | Flip | 25 | 26.1 | 48.7 | 49.0 | 47.3 | 31.8 | 22.4 | 21.7 |
| | Mean | | 35.6 | 38.3 | 37.9 | 37.4 | 37.4 | 24.4 | **23.3** |
| IMDB | Uniform | 5 | 36.9 | 26.7 | 27.9 | 27.6 | 35.5 | 25.4 | 25.0 |
| | Uniform | 10 | 26.2 | 25.8 | 27.2 | 26.1 | 24.9 | 23.3 | 22.3 |
| | Uniform | 25 | 22.2 | 21.4 | 23.0 | 20.1 | 21.0 | 18.9 | 18.7 |
| | Flip | 5 | 36.9 | 49.2 | 49.2 | 49.2 | 41.8 | 25.8 | 25.2 |
| | Flip | 10 | 26.2 | 47.8 | 48.3 | 47.5 | 28.0 | 22.1 | 22.0 |
| | Flip | 25 | 22.2 | 39.4 | 39.6 | 36.6 | 23.5 | 19.2 | 18.5 |
| | Mean | | 28.5 | 35.0 | 35.9 | 34.5 | 29.1 | 22.5 | **22.0** |
| Twitter | Uniform | 5 | 35.9 | 37.1 | 51.7 | 44.1 | 32.0 | 41.5 | 31.0 |
| | Uniform | 10 | 23.6 | 33.5 | 49.5 | 40.2 | 22.2 | 33.6 | 22.3 |
| | Uniform | 25 | 16.3 | 25.5 | 40.6 | 26.4 | 16.6 | 20.0 | 15.5 |
| | Flip | 5 | 35.9 | 56.2 | 61.6 | 54.8 | 36.4 | 23.4 | 15.8 |
| | Flip | 10 | 23.6 | 53.8 | 59.0 | 48.9 | 26.1 | 15.9 | 12.9 |
| | Flip | 25 | 16.3 | 43.0 | 52.5 | 36.7 | 20.5 | 13.3 | 12.8 |
| | Mean | | 25.3 | 41.5 | 52.5 | 41.9 | 25.7 | 24.6 | **18.4** |

Table 2: NLP dataset results. Percent trusted is the trusted fraction multiplied by 100. Unless otherwise indicated, all values are percentages representing the area under the error curve computed at 11 test points. The best mean result is bolded.

take a convex combination of an identity matrix and the matrix $11^{\mathsf{T}}/K$. We refer to the coefficient of $11^{\mathsf{T}}/K$ as the corruption strength for a "uniform" corruption. A "flip" corruption at strength $m$ involves, for each row, giving an off-diagonal column probability mass $m$ and the entries along the diagonal probability mass $1 - m$. Finally, a more realistic corruption is hierarchical corruption. For this corruption, we apply uniform corruption only to semantically similar classes; for example, "bed" may be corrupted to "couch" but not "beaver" in CIFAR-100. For CIFAR-100, examples are deemed semantically similar if they share the same "superclass" label specified by the dataset creators.

**Experiments and Analysis of Results.** We train the models described in Section 4.1 under uniform, label-flipping, and hierarchical label corruptions at various fractions of trusted data. To assess the performance of the GLC, we compare it to other loss correction methods and two baselines: one where we train a network only on trusted data without any label corrections, and one where the network trains on all data without any label corrections. We record errors on the test sets at the corruption strengths $\{0, 0.1, \ldots, 1.0\}$. Since we compute the model's accuracy at numerous corruption strengths, CIFAR experiments involve training over 500 Wide Residual Networks. In Tables 1 and 2, we report the *area under the error curves* across corruption strengths $\{0, 0.1, \ldots, 1.0\}$ for all baselines and corrections. A sample of error curves are displayed in Figure 2. These curves are the linear interpolation of the errors at the eleven corruption strengths.

Across all experiments, the GLC obtains better area under the error curve than the baselines and the Forward and Distillation methods. The rankings of the other methods and baselines are mixed. On MNIST, training on the trusted data alone outperforms all methods save for the GLC and Confusion Matrix, but performs significantly worse on CIFAR-100, even with large trusted fractions.

The Confusion Matrix correction performs second to the GLC, which indicates that normalized confusion matrices are not as accurate as our method of estimating $C$. We verified this by inspecting the estimates directly, and found that normalized confusion matrices give a highly biased estimate due to taking an argmax over class scores rather than using random sampling. Figure 1 shows an example of this bias in the case of label flipping corruption at a strength of $7/10$.

Interestingly, Forward Gold performs worse than Forward on several datasets. We did not observe the same behavior when turning off the corresponding component of the GLC, and believe it may be due to variance introduced during training by the difference in signal provided by the Forward method's

$C$ estimate and the clean labels. The GLC provides a superior $C$ estimate, and thus may be better able to leverage training on the clean labels. Additional results on SVHN are in the Supplementary Material.

We also compare the GLC to the recent work of Ren et al. [21], which proposes a loss correction that uses a trusted set and meta-learning. We find that the GLC consistently outperforms this method. To conserve space, results are in the Supplementary Material.

|  | Percent Trusted | Trusted Only | No Corr. | Forward | Forward Gold | Distill. | Confusion Matrix | GLC (Ours) |
|---|---|---|---|---|---|---|---|---|
| CIFAR-10 | 1 | 62.9 | 28.3 | 28.1 | 30.9 | 60.4 | 31.9 | 26.9 |
|  | 5 | 39.6 | 27.1 | 26.6 | 25.5 | 28.1 | 27 | 21.9 |
|  | 10 | 31.3 | 25.9 | 25.1 | 22.9 | 17.8 | 24.2 | 19.2 |
| Mean |  | 44.6 | 27.1 | 26.6 | 26.4 | 35.44 | 27.7 | **22.7** |
| CIFAR-100 | 5 | 82.4 | 71.1 | 73.9 | 73.6 | 88.3 | 74.1 | 68.7 |
|  | 10 | 67.3 | 66 | 68.2 | 66.1 | 62.5 | 63.8 | 56.6 |
|  | 25 | 52.2 | 56.9 | 56.9 | 51.4 | 39.7 | 50.8 | 40.8 |
| Mean |  | 67.3 | 64.7 | 66.3 | 63.7 | 63.5 | 62.9 | **55.4** |

Table 3: Results when obtaining noisy labels by sampling from the softmax distribution of a weak classifier. Percent trusted is the trusted fraction multiplied by 100. Unless otherwise indicated, all values are the percent error. The best average result for each dataset is shown in bold.

## 4.4 Weak Classifier Labels

Our next benchmark for the GLC is to use noisy labels obtained from a weak classifier. This models the scenario of label noise arising from a classification system weaker than one's own, but with access to information about the true labels that one wishes to transfer to one's own system. For example, scraping image labels from surrounding text on web pages provides a valuable signal, but these labels would train a sub-par classifier without correcting the label noise. This setting exactly satisfies the conditional independence assumption on label noise used for our $\widehat{C}$ estimate, because the weak classifier does not take the true label as input when outputting noisy labels.

**Weak Classifier Label Generation.** To obtain the labels, we train 40-layer Wide Residual Networks on CIFAR-10 and CIFAR-100 with clean labels for ten epochs each. Then, we sample from their softmax distributions with a temperature of 5, and fix the resulting labels. This results in noisy labels which we use in place of the labels obtained through the uniform, flip, and hierarchical corruption methods. The labelings produced by the weak classifiers have accuracies of $40\%$ on CIFAR-10 and $7\%$ on CIFAR-100. Despite the presence of highly corrupted labels, we are able to significantly recover performance with the use of a trusted set. Note that unlike the previous corruption methods, weak classifier labels have only one corruption strength. Thus, performance is measured in percent error rather than area under the error curve. Results are displayed in Table 3.

**Analysis of Results.** On average, the GLC outperforms all other methods in the weak classifier label experiments. The Distillation method performs better than the GLC by a small margin at the highest trusted fraction, but performs worse at lower trusted fractions, indicating that the GLC enjoys superior data efficiency. This is highlighted by the GLC attaining a $26.94\%$ error rate on CIFAR-10 with a trusted fraction of only $1\%$, down from the original error rate of $60\%$. It should be noted, however, that training with no correction attains $28.32\%$ error on this experiment, suggesting that the weak classifier labels have low bias. The improvement conferred by the GLC is greater with larger trusted fractions.

## 5 Discussion

**Data Efficiency.** We have seen that the GLC works for small trusted fractions. We further corroborate its data efficiency by turning to the Clothing1M dataset [27]. Clothing1M is a massive dataset with both human-annotated and noisy labels, which we use to compare the data efficiency of the GLC to that of Distillation when very few trusted labels are present. It consists in 1 million noisily labeled clothing images obtained by crawling online marketplaces. 50,000 images have human-annotated examples, from which we take subsamples as our trusted set.

For both the GLC and Distillation, we first fine-tune a ResNet-34 on untrusted training examples for four epochs, and use this to estimate our corruption matrix. Thereafter, we fine-tune the network for

four more epochs on the combined trusted and untrusted sets using the respective method. During fine tuning, we freeze the first seven layers, and train using gradient descent with Nesterov momentum and a cosine learning rate schedule. For preprocessing, we randomly crop and use mirroring. We also upsample the trusted dataset, finding this to give better performance for both methods.

As shown in Figure 3, the GLC outperforms Distillation by a large margin, especially with fewer trusted examples. This is because Distillation requires fine-tuning a classifier on the trusted data alone, which generalizes poorly with very few examples. By contrast, estimating the $C$ matrix can be done with very few examples. Correspondingly, we find that our advantage decreases as the number of trusted examples increases.

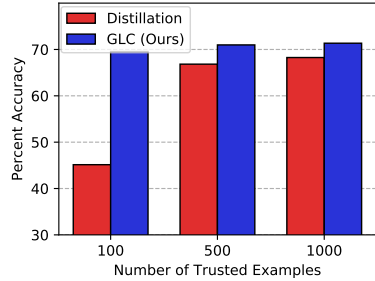

Figure 3: Data efficiency of our method compared to Distillation on Clothing1M.

With more trusted labels, performance on Clothing1M saturates, as evident in Figure 3. We consider the extreme and train on the entire trusted set for Clothing1M. We fine-tune a pre-trained 50-layer ResNeXt [28] on untrusted training examples to estimate our corruption matrix. Then, we fine-tune the ResNeXt on all training examples. During fine-tuning, we use gradient descent with Nesterov momentum. During the first two epochs, we tune only the output layer with a learning rate of $10^{-2}$. Thereafter, we tune the whole network at a learning rate of $10^{-3}$ for two epochs, and for another two epochs at $10^{-4}$. Then we apply our loss correction. Now, we fine-tune the entire network at a learning rate of $10^{-3}$ for two epochs, continue training at $10^{-4}$, and early-stop based upon the validation set. In a previous work, Xiao et al. [27] obtain $78.24\%$ in this setting. However, our method obtains a state-of-the-art accuracy of $80.67\%$, while with this procedure the Forward method only obtains $79.03\%$ accuracy.

**Improving $\widehat{C}$ Estimation.** For some datasets, the classifier $\widehat{p}(\widetilde{y} \mid x)$ may be a poor estimate of $p(\widetilde{y} \mid x)$, presenting a bottleneck in the estimation of $\widehat{C}$ for the GLC. To see the extent to which this could impact performance, and whether simple methods for improving $\widehat{p}(\widetilde{y} \mid x)$ could help, we ran several variants of the GLC experiment on CIFAR-100 under the label flipping corruption at a trusted fraction of $5/100$ which we now describe. For all variants, we averaged the area under the error curve over five random initializations.

1. In the first variant, we replaced the GLC estimate of $\widehat{C}$ with $C$, the true corruption matrix.
2. As demonstrated by Hendrycks and Gimpel [6] and Guo et al. [5], modern deep neural network classifiers tend to have overconfident softmax distributions. We found this to be the case with our $\widehat{p}(\widetilde{y} \mid x)$ estimate, despite the higher entropy of the noisy labels, so we used the temperature scaling confidence calibration method proposed in the paper to calibrate $\widehat{p}(\widetilde{y} \mid x)$.
3. Suppose we know the base rates of corrupted labels $\widetilde{b}$, where $\widetilde{b}_i = p(\widetilde{y} = i)$, and the base rate of true labels $b$ of the trusted set. If we posit that $\widehat{C}_0$ corrupted the labels, then we should have $b^\mathsf{T}\widehat{C}_0 = \widetilde{b}^\mathsf{T}$. Thus, we may obtain a superior estimate of the corruption matrix by computing a new estimate $\widehat{C} = \operatorname{argmin}_{\widehat{C}} \|b^\mathsf{T}\widehat{C}_0 - \widetilde{b}^\mathsf{T}\|_2^2 + \lambda\|\widehat{C} - \widehat{C}_0\|_2^2$ subject to $\widehat{C}1 = 1$.

We found that using the true corruption matrix as our $\widehat{C}$ provides a benefit of $0.96$ percentage points in area under the error curve, but neither the confidence calibration nor the base rate incorporation was able to change the performance from the original GLC. This indicates that the GLC is robust to the use of uncalibrated networks for estimating $C$, and that improving its performance may be difficult without directly improving the performance of the neural network used to estimate $\widehat{p}(y \mid x)$.

## 6 Conclusion

In this work, we have shown the impact of having a small set of trusted examples on label noise robustness in neural network classifiers. We proposed the Gold Loss Correction (GLC), a method for coping with label noise. This method leverages the assumption that the model has access to a small set of correct labels in order to yield accurate estimates of the noise distribution. Throughout our experiments, the GLC surpasses previous label noise robustness methods across various natural language processing and vision domains which we showed by considering several corruptions and numerous strengths, including severe strengths. These results demonstrate that the GLC is a powerful, data-efficient method for improving robustness to label noise.

**Acknowledgments**

We thank NVIDIA for donating GPUs used in this research.

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
