[Supplementary Material]

# A  Proof: Separability Implies $Y \perp \widetilde{Y} \mid X$

We show here that the conditional independence assumption required by our $\widehat{C}$ estimator is satisfied when the data are separable, meaning that the label is deterministic given the input.

Let $X, Y, \widetilde{Y}$ be random variables following a data distribution $P_{X,Y,\widetilde{Y}}$, where $Y$ and $\widetilde{Y}$ are categorical. Semantically, $Y$ represents the true label, and $\widetilde{Y}$ represents the noisy label. Suppose that the data are separable, meaning that $P_{Y|X}(y \mid x) = 0$ holds for all $y$ but $y^*$, in which case we have $P_{Y|X}(y^* \mid x) = 1$. For brevity in the rest of the proof, we will use shorthand probability notation, i.e. $p(y^* \mid x) = 1$. Using the separability assumption, we have

$$p(\widetilde{y} \mid x) = \sum_y p(y, \widetilde{y} \mid x) = \sum_y p(\widetilde{y} \mid y, x)p(y \mid x) = p(\widetilde{y} \mid y^*, x). \tag{1}$$

We will use this to show that $p(y, \widetilde{y} \mid x) = p(y \mid x)p(\widetilde{y} \mid x)$ for all $y, \widetilde{y}, x$. Let $\widetilde{y}$ and $x$ be given. For $y \neq y^*$, we have

$$p(y, \widetilde{y} \mid x) = \frac{p(y, \widetilde{y}, x)}{p(x)} = \frac{0}{p(x)} = 0,$$

because separability implies $p(y, x) = 0$ for $y \neq y^*$. This is also equal to $p(y \mid x)p(\widetilde{y} \mid x)$, so the case where $y \neq y^*$ is covered. Suppose $y = y^*$. We have

$$p(y^*, \widetilde{y} \mid x) = \frac{p(y^*, \widetilde{y}, x)}{p(x)} = \frac{p(\widetilde{y} \mid y^*, x)p(y^*, x)}{p(x)} = p(\widetilde{y} \mid x)p(y^* \mid x),$$

where in the last step we use equation (1). This completes the proof.

# B    Additional Results and Error Plots

|  | Corruption Type | Percent Trusted | Trusted Only | No Corr. | Ren et al. | GLC (Ours) |
|---|---|---|---|---|---|---|
| **MNIST** | Uniform | 5 | 37.6 | 12.9 | 10.6 | 10.3 |
|  | Uniform | 10 | 12.9 | 12.3 | 7.7 | 6.3 |
|  | Uniform | 25 | 6.6 | 9.3 | 8.5 | 4.7 |
|  | Flip | 5 | 37.6 | 50.1 | 20.2 | 3.4 |
|  | Flip | 10 | 12.9 | 51.1 | 22.7 | 2.9 |
|  | Flip | 25 | 6.6 | 47.7 | 22.7 | 2.6 |
|  | Mean | | 19.0 | 30.6 | 15.4 | **5.0** |
| **CIFAR-10** | Uniform | 5 | 39.6 | 31.9 | 30.5 | 9.0 |
|  | Uniform | 10 | 31.3 | 31.9 | 30.8 | 6.9 |
|  | Uniform | 25 | 17.4 | 32.7 | 33.3 | 6.4 |
|  | Flip | 5 | 39.6 | 53.3 | 21.9 | 6.6 |
|  | Flip | 10 | 31.3 | 53.2 | 23.0 | 6.2 |
|  | Flip | 25 | 17.4 | 52.7 | 24.4 | 6.1 |
|  | Mean | | 29.4 | 42.6 | 27.3 | **6.9** |
| **CIFAR-100** | Uniform | 5 | 82.4 | 48.8 | 68.5 | 42.4 |
|  | Uniform | 10 | 67.3 | 48.4 | 71.5 | 33.9 |
|  | Uniform | 25 | 52.2 | 45.4 | 72.8 | 27.3 |
|  | Flip | 5 | 82.4 | 62.1 | 67.2 | 27.1 |
|  | Flip | 10 | 67.3 | 61.9 | 68.4 | 25.8 |
|  | Flip | 25 | 52.2 | 59.6 | 71.5 | 24.7 |
|  | Mean | | 67.3 | 54.4 | 70.0 | **30.2** |

Table 1: Results on the method of Ren et al. [19]. Results from all methods besides Ren et al. are copied from Table 2. Percent trusted is the trusted fraction multiplied by 100. Unless otherwise indicated, all values are percentages representing the area under the error curve computed at 11 test points. The best mean result is shown in bold.

| | Corruption Type | Percent Trusted | Trusted Only | No Corr. | Forward | Forward Gold | Distill. | Confusion Matrix | GLC (Ours) |
|---|---|---|---|---|---|---|---|---|---|
| **MNIST** | Uniform | 5 | 37.6 | 12.9 | 14.5 | 13.5 | 42.1 | 21.8 | 10.3 |
| | Uniform | 10 | 12.9 | 12.3 | 13.9 | 12.3 | 9.2 | 15.1 | 6.3 |
| | Uniform | 25 | 6.6 | 9.3 | 11.8 | 9.2 | 5.8 | 11.0 | 4.7 |
| | Flip | 5 | 37.6 | 50.1 | 51.7 | 41.4 | 46.6 | 11.7 | 3.4 |
| | Flip | 10 | 12.9 | 51.1 | 48.8 | 36.4 | 32.4 | 5.6 | 2.9 |
| | Flip | 25 | 6.6 | 47.7 | 50.2 | 37.1 | 28.2 | 3.8 | 2.6 |
| | Mean | | 19.0 | 30.6 | 31.8 | 25.0 | 27.4 | 11.5 | **5.0** |
| **SVHN** | Uniform | 0.1 | 80.4 | 25.5 | 26.2 | 26.8 | 80.9 | 25.7 | 24.4 |
| | Uniform | 1 | 79.7 | 25.5 | 24.2 | 24.9 | 80.4 | 28.2 | 28.1 |
| | Uniform | 5 | 24.3 | 25.5 | 15.0 | 15.7 | 24.1 | 2.7 | 2.8 |
| | Flip | 0.1 | 80.4 | 51.0 | 51.0 | 50.9 | 89.1 | 19.8 | 19.4 |
| | Flip | 1 | 79.7 | 51.0 | 43.9 | 49.5 | 86.3 | 17.8 | 21.7 |
| | Flip | 5 | 24.3 | 51.0 | 43.2 | 49.0 | 17.6 | 2.2 | 2.2 |
| | Mean | | 61.5 | 38.2 | 33.9 | 36.1 | 63.1 | **16.1** | 16.4 |
| **CIFAR-10** | Uniform | 5 | 39.6 | 31.9 | 9.1 | 27.9 | 29.7 | 22.4 | 9.0 |
| | Uniform | 10 | 31.3 | 31.9 | 8.6 | 20.6 | 18.3 | 22.7 | 6.9 |
| | Uniform | 25 | 17.4 | 32.7 | 7.7 | 27.1 | 11.6 | 16.7 | 6.4 |
| | Flip | 5 | 39.6 | 53.3 | 38.6 | 47.8 | 29.7 | 8.1 | 6.6 |
| | Flip | 10 | 31.3 | 53.2 | 36.5 | 51.0 | 18.1 | 8.2 | 6.2 |
| | Flip | 25 | 17.4 | 52.7 | 37.6 | 49.5 | 11.8 | 7.1 | 6.1 |
| | Mean | | 29.4 | 42.6 | 23.0 | 37.3 | 19.9 | 14.2 | **6.9** |
| **CIFAR-100** | Uniform | 5 | 82.4 | 48.8 | 47.7 | 49.6 | 87.5 | 53.6 | 42.4 |
| | Uniform | 10 | 67.3 | 48.4 | 47.2 | 48.9 | 61.2 | 49.7 | 33.9 |
| | Uniform | 25 | 52.2 | 45.4 | 43.6 | 46.0 | 39.8 | 39.6 | 27.3 |
| | Flip | 5 | 82.4 | 62.1 | 61.6 | 62.6 | 87.1 | 28.6 | 27.1 |
| | Flip | 10 | 67.3 | 61.9 | 61.0 | 62.2 | 61.9 | 26.9 | 25.8 |
| | Flip | 25 | 52.2 | 59.6 | 57.5 | 61.4 | 40.0 | 25.1 | 24.7 |
| | Hierarchical | 5 | 82.4 | 50.9 | 51.0 | 52.4 | 87.1 | 45.8 | 34.8 |
| | Hierarchical | 10 | 67.3 | 51.9 | 50.5 | 52.1 | 61.7 | 38.8 | 30.2 |
| | Hierarchical | 25 | 52.2 | 54.3 | 47.0 | 51.1 | 39.7 | 29.7 | 25.4 |
| | Mean | | 67.3 | 53.7 | 51.9 | 54.0 | 62.9 | 37.5 | **30.2** |

Table 2: Vision dataset results. These differ from the results in the paper by the addition of SVHN. Percent trusted is the trusted fraction multiplied by 100. Unless otherwise indicated, all values are percentages representing the area under the error curve computed at 11 test points. The best mean result is shown in bold.

CIFAR-100, Uniform, 5% trusted
CIFAR-100, Uniform, 10% trusted
CIFAR-100, Uniform, 25% trusted

CIFAR-100, Flip, 5% trusted
CIFAR-100, Flip, 10% trusted
CIFAR-100, Flip, 25% trusted

CIFAR-100, Hierarchical, 5% trusted
CIFAR-100, Hierarchical, 10% trusted
CIFAR-100, Hierarchical, 25% trusted

CIFAR-10, Uniform, 5% trusted
CIFAR-10, Uniform, 10% trusted
CIFAR-10, Uniform, 25% trusted

CIFAR-10, Flip, 5% trusted
CIFAR-10, Flip, 10% trusted
CIFAR-10, Flip, 25% trusted

Legend: Ours, Confusion, Distillation, Forward Gold, Forward, No Correction

Axes: Test Error vs Corruption Strength

Figure 2: Error curves for numerous label correction methods on vision datasets using several label corruption types and a full range of label corruption strengths.

Figure 4: Error curves for numerous label correction methods on NLP datasets using several label corruption types and a full range of label corruption strengths.