[Reviews · NeurIPS 2018]

Reviewer 1



Summary A method for learning a classifier robust to label noise is proposed. In contrast with most previous work, the authors take a simplifying but realistic assumption: there is of a small subset of the training set that has been sanitized (where labels are not noisy). Leveraging prior work, a noise transition matrix is first estimated and then used to correct the loss function for classification. Very extensive experiments support the validity of the method. Detailed comments This is a good contribution, although rather incremental with respect to Patrini et al. '17. Experimentation is extensive and the presentation is mostly fluid and clear. I found few passages a little confusing and encourage the authors to elaborate better: * line 102: ’This has the effect of allowing the GLC to handle a degree of instance-dependency in the label noise’ Why? * line 115: ‘However, we find that the GLC still works well in practice,;perhaps because this assumption is also satisfied when the data are separable, in the sense that each x, has a single true y, which is approximately true in many of our experiments’ ? I also did not understand what is the Forward Gold method that is compared with. In particular, what do you mean with identity on trusted examples? I can see the use of the Forward correction with your estimator of C in page 2 (please add numbers to formulae). But this is precisely your proposed method GLC, correct? Another question on the experiment: why does the proposed method not exploit the clean label for training of the final network? If they are assumed to be available, why not using them during training? This does not sound realistic. You can probably improve further performance of your method utilising the clean data during training, not just for the correction but also as training points. Is this done? On the experiments on Clothing1M, ~ line 279: the best result in [23] was obtained with an AlexNet and therefore is not clearly comparable. [17] reports accuracy of 80.38, using a simpler 50-ResNet. Please clarify what you mean with state of the art exactly. Minor: * Table in appendix: best results are *not* given in bold * Do you have a citation for the cosine learning schedule?

Reviewer 2



The submission describes a training methodology when you are given a dataset partitioned into a trusted and untrusted dataset. In the abstract and introduction, the authors justify the need for such a method and give a brief overview of the related work. They call their method ‘Gold Loss Correction’ (GLC). The algorithm describes approximating the probability of an untrusted label given an input by training some classifier with softmax output and then using this model alongside the trusted dataset to construct a matrix of corruption probabilities. They do this by approximating the corruption probabilities by simply measuring the likelihood a model will classify some class from the partition of the dataset that is without doubt of some true class. Ofcourse this is non-normalized, so they mention trying both a normalized (The corruption matrix becomes a confusion matrix) and the non-normalized version. In either case they use the matrix to weight the output of a newly trained classifier to consider this likelihood. The results show that the normalized version performs slightly worse, and so GLC requires no normalization. They compare this approach to other Training procedures that account for untrusted data, but only one other that utilizes a trusted set. The comparison to Forward Loss Correction has this inherent bias because it does not utilize the important information that is the foundation of this work. They attempt to remedy this by adjusting the Forward Loss Corrections corruption matrix with the identity on the trusted component of each dataset. This ended up not assisting, which they hypothesize is due to variance introduced in their setup. This specific differential doesn’t seem clear, and I feel further investigation into why the adjustment had little effect may shed light onto strengths and weaknesses of their algorithm. The results show a strong advantage to using GLC over the other algorithms in most cases (except for one against distillation in a niche setting) as well as it being robust and data-efficient. Though it should be mentioned that GLC works on a baseline assumption that the trusted and untrusted labels are conditionally independent on x. They backup this assumption by saying most systems that induce noise usually do not know the true label which is valid, but it would be interesting to see the comparative results in cases where this assumption is broken in examples like worst-case adversarial settings. The paper is well-written, and they describe both the related work and their procedures in detail. The concept underlying their innovative training methodology is not purely original because they are using the same foundation as in Patrini et al 2017, but then originally extend it to utilize trusted component of the dataset.

Reviewer 3



This paper proposes a method to train deep networks using corrupted training data. The core idea is to exploit a small set of clean labels from a trusted source. The authors argue that getting such additional data is practical, which is reasonable. The paper is well written and the experiments are quite thorough. My main concern with the paper is regarding its novelty. Training deep models using noisy labels has been previous explored. The main contribution of the paper is in estimating the corruption matrix using a rather straightforward process as described in Sec 3.1. I feel this is not sufficient for acceptance in a venue like NIPS.